# Adaptive Changes in the Central Control of Energy Homeostasis Occur in Response to Variations in Energy Status

**DOI:** 10.3390/ijms22052728

**Published:** 2021-03-08

**Authors:** Cassandra Gastelum, Lynnea Perez, Jennifer Hernandez, Nikki Le, Isabella Vahrson, Sarah Sayers, Edward J. Wagner

**Affiliations:** 1Graduate College of Biomedical Sciences, Western University of Health Sciences, Pomona, CA 91766, USA; cassandra.gastelum@westernu.edu (C.G.); lynnea.perez@westernu.edu (L.P.); jenniferhernandez@westernu.edu (J.H.); nikki.le@westernu.edu (N.L.); isabella.vahrson@westernu.edu (I.V.); sarah.sayers@westernu.edu (S.S.); 2College of Osteopathic Medicine of the Pacific, Western University of Health Sciences, Pomona, CA 91766, USA

**Keywords:** sex difference, estradiol, nociceptin/orphanin FQ, pituitary adenylate cyclase-activating polypeptide, obesity, fasting

## Abstract

Energy homeostasis is regulated in coordinate fashion by the brain-gut axis, the homeostatic energy balance circuitry in the hypothalamus and the hedonic energy balance circuitry comprising the mesolimbcortical A_10_ dopamine pathway. Collectively, these systems convey and integrate information regarding nutrient status and the rewarding properties of ingested food, and formulate it into a behavioral response that attempts to balance fluctuations in consumption and food-seeking behavior. In this review we start with a functional overview of the homeostatic and hedonic energy balance circuitries; identifying the salient neural, hormonal and humoral components involved. We then delve into how the function of these circuits differs in males and females. Finally, we turn our attention to the ever-emerging roles of nociceptin/orphanin FQ (N/OFQ) and pituitary adenylate cyclase-activating polypeptide (PACAP)—two neuropeptides that have garnered increased recognition for their regulatory impact in energy homeostasis—to further probe how the imposed regulation of energy balance circuitry by these peptides is affected by sex and altered under positive (e.g., obesity) and negative (e.g., fasting) energy balance states. It is hoped that this work will impart a newfound appreciation for the intricate regulatory processes that govern energy homeostasis, as well as how recent insights into the N/OFQ and PACAP systems can be leveraged in the treatment of conditions ranging from obesity to anorexia.

## 1. The Hypothalamic Energy Balance Circuit in Homeostatic Feeding

Energy homeostasis, the intricate balance between energy intake and expenditure, is regulated in coordinate fashion by homeostatic and hedonic neural circuits [1]. Aberrations in these circuits are implicated in the pathophysiology of conditions such as obesity, type-II diabetes, and food addiction [2,3,4,5]. Homeostatic control of energy balance is attributed to the hypothalamic energy balance neural circuitry, which integrates information relayed from the brainstem regarding the energy/nutritional status of the organism, based on chemical and mechanical cues from the gastrointestinal (GI) tract [6,7]. These communications are ultimately encoded by various hypothalamic nuclei and their associated neuronal populations, including those in the arcuate nucleus (ARC), ventromedial nucleus (VMN), lateral hypothalamus (LH), dorsomedial nucleus (DMN), and the paraventricular nucleus (PVN); producing orexigenic and anorexigenic signals to, respectively, stimulate and suppress energy intake as well as altering energy expenditure [1,8].

Excitatory input from steroidogenic factor (SF-1) neurons, located in the VMN, impinging on proopiomelanocortin (POMC) neurons in the ARC represent a critical anorexigenic synapse in homeostatic energy balance that, when activated, suppresses energy intake and enhances energy expenditure [5,8,9,10,11,12]. SF-1 is a transcription factor encoded by the NR5A1 gene, and activation of these neurons in the VMN leads to glutamatergic stimulation of POMC neurons [2,5,8,9,10,11]. Activation of POMC expressing neurons within the hypothalamic melanocortin system ultimately leads to formation of POMC posttranslational cleavage products such as α-melanocortin stimulating hormone (α-MSH), which following release from axon terminals go on to bind downstream effectors like melanocortin 4 receptors (MC4R) expressed on corticotropin-releasing hormone (CRH) neurons in the PVN [13,14,15,16,17]. In line with α-MSH functioning as a satiety mediator, the percentage of α-MSH neurons colocalizing with c-Fos in the ARC is greatest at the end of a meal, compared to the beginning of or hours after consumption [18]. In addition to α-MSH, POMC neurons also release β-endorphin (following posttranslational modification), and co-express cocaine- and amphetamine- regulated transcript (CART) [19,20]. Overall, POMC signaling reduces food intake, increases energy expenditure, and regulates glucose metabolism [16,17,19,21]. Prevailing glucose concentrations play a key role in relaying nutrition/energy state cues to the homeostatic energy balance circuity, where anorexigenic and orexigenic ARC neurons have glucose sensing abilities. ARC POMC/CART neurons are categorized as glucose-responsive neurons and take up glucose via a glucose transporter (GLUT2) where it then metabolizes, producing ATP and thereby promoting the closure of ATP-dependent potassium (K_ATP_) channels to reduce the outflow of K^+^, ultimately leading to the depolarization of the cell [22,23]. This aligns POMC cellular excitability and firing rate in direct proportion to glucose concentrations, with satiety signaling accentuated as glucose levels rise (e.g., during or shortly following a meal) [22,23]. It should therefore not be surprising that perturbations in signaling at this VMN SF-1/ARC POMC synapse can pose detrimental consequences for energy balance, where null mutations in POMC, its cleavage enzymes or downstream receptors, as well as lesioning in the VMN, ultimately leads to hyperphagia and obese phenotypes in rodents and humans [16,17,24].

At the opposing end of hypothalamic energy balance control are neuronal populations that promote orexigenic or appetite-stimulating effects. Alongside POMC neurons, the ARC houses neuropeptide Y (NPY) and agouti-related peptide (AgRP) co-expressing neurons, as well as ghrelin-containing somata [25,26,27]. Activation of this subset of neurons pleiotropically dampens the aforementioned anorexigenic signaling. For example, following NYP/AgRP neuronal activation, direct inhibition of neighboring POMC neurons is achieved via synaptic release of the inhibitory amino acid neurotransmitter γ-aminobutyric acid (GABA) on POMC soma [28]. Further downstream modulation of POMC signaling is mediated by AgRP antagonism on MC4R or via NPY acting on various receptor subtypes, ultimately impeding the anorexigenic signaling of α-MSH to induce feeding and reduce energy expenditure [15,23]. Additionally, NPY/AgRP neurons in the ARC are glucose-sensitive neurons and their activity/firing rate is inversely proportional to ambient glucose levels [23]. Neurons found within the LH also relay orexigenic signals, which are mediated by neurotransmission of melanin-concentrating hormone (MCH) neurons and orexin neurons [29,30]. Indeed, orexin has been shown to electrically silence POMC neurons by enhancing GABAergic and diminishing glutamatergic inputs onto these cells [31]. In addition, endocannabinoids elicit their orexigenic effects in part through retrograde inhibition of GABAergic inputs onto MCH neurons [32].

In addition to the ascending inputs from the brainstem, the neurons comprising the hypothalamic energy balance circuit are also susceptible to the influence of circulating peripheral hormones—leptin, ghrelin, insulin and sex hormones. Sensitivity to these transient peripheral hormones (and other periphery signaling molecules) is especially true for ARC neuronal populations, as this region lies in close proximity to the third ventricle (3V) and median eminence; a circumventricular organ with an incomplete blood–brain barrier [33,34]. Therefore, the ARC is in direct contact with systemic circulating hormones, permitting critical communication about the energy/nutritional status of the body to ARC neurons that promotes energy homeostasis. Leptin is synthesized by white adipose tissue (WAT), with levels fluctuating in proportion to fat mass. Leptin acts as a potent suppressor of food intake, while also stimulating metabolism and reducing excessive stored energy [35]. POMC and SF-1 neurons are depolarized by leptin receptor (LEPR) activation via JAK/signal-transducer-and-activator-of-transcription (STAT) and phosphatidylinositol-4,5-bisphosphate 3-kinase (PI3K) pathway signaling [36,37,38,39], while LEPR activity on AgRP neurons activates ATP-gated K^+^ (K_ATP_) channels leading to potassium outflow, hyperpolarization and decreases in firing [40,41,42]. The anorexigenic effect of leptin also involves a reduction in hypothalamic endocannabinoid levels [43]. Conversely, hypothalamic levels of endocannabinoids are increased in leptin receptor-deficient *fa/fa* rats and leptin-deficient *ob/ob* mice [43]. Similar to leptin, insulin signaling in the hypothalamus promotes an anorexigenic tone and elicits transient receptor potential (TRP)C5 channel-induced excitation of POMC neurons following receptor activation [44]. While for insulin, several prior studies reported inhibitory responses in POMC neurons due to activation of K_ATP_ channels, it is now known that the proportion of excitatory vs. inhibitory insulin-induced responses is dependent on ambient levels of protein tyrosine phosphatase 1B (PTP1B) and T-cell protein tyrosine phosphatase (TCPTP) activity [45,46,47]. Both enzymes are expressed in ARC neurons integral to the regulation of energy balance. PTP1B and TCPTP are key regulators of cell metabolism, as PTP1B decreases leptin activity and TCPTP attenuates insulin signaling via dephosphorylation of JAK2 tyrosine kinase and the insulin receptor, respectively, in POMC neurons, whereas in NPY/AgRP neurons TCPTP attenuates insulin but not leptin signaling [48,49,50]. Deletion of both enzymes in POMC neurons, or of TCPTP in NPY/AgRP neurons, from obese mice results in promoted weight loss due to decreased food consumption and increased WAT browning along with elevated uncoupling protein (UCP)-1 expression due to enhanced leptin and insulin signaling [48,49].

Conversely, levels of the orexigenic gut-derived peptide ghrelin increase in response to negative energy balance. As a result, the peptide binds to receptors on ARC NPY/AgRP neurons to stimulate these neurons, and thereby promote feeding behavior and energy storage [35]. The orexigenic effect of ghrelin also depends on enhanced hypothalamic production of endocannabinoids and activation of the energy-sensing signaling molecule AMP-dependent protein kinase (AMPK) that, in turn, elicits retrograde inhibition of excitatory input impinging on parvocellular neurons in the PVN [51]. These actions require functionally intact cannabinoid CB1 and ghrelin receptor systems [51,52]. AMPK functions as an important signaling molecule within the hypothalamic energy balance circuitry, and exists as a heterotrimeric complex comprising α-, β-, and γ-subunits [53,54,55,56]. The α subunit is the catalytic subunit, while the β and γ subunits are involved in glycogen and AMP/ATP binding, respectively [53,54,55,56]. AMPK activity is stimulated by phosphorylation via two upstream kinases, human tumor suppressor LKB1 or Ca^2+^/calmodulin-dependent protein kinase kinase-β (CaMKKβ) that is triggered by cellular stress, cytokines, hormones like those mentioned above, as well as by increases in AMP/ATP ratio [53,54,55,56]. This in turn inhibits anabolic pathways and activates catabolic pathways to generate ATP [53,54,55,56]. Leptin, insulin, α-MSH, high plasma glucose concentrations and refeeding all inhibit AMPK whereas AgRP increases it [57]. On the other hand, AMPK activation is necessary for leptin to inhibit fatty acid synthesis and thereby promote fatty acid oxidation in skeletal muscle via phosphorylation of acetyl coenzyme A carboxylase [58]. This is consistent with other examples of opposing cannabinoid- and ghrelin-induced effects on AMPK activity in the central nervous system (CNS) versus the periphery [59]. When the effects of AMPK are constitutively manifest in transgenic mice bearing a gain-of-function mutation in the γ2 subunit, this brings about ghrelin-dependent hyperphagia that leads to obesity, glucose intolerance and hyperinsulinemia [60]. The dynamic interplay between the aforementioned peripheral hormones and the homeostatic energy balance circuitry is graphically depicted in Figure 1.

## 2. The Mesolimbic Dopamine Network and Hedonic Feeding Behavior

Alongside the homeostatic-hypothalamic circuitry, energy balance is also modulated by hedonic aspects of feeding behavior pertaining to reward-based food intake, or eating for pleasure. Amounting evidence in rodents and humans now support the theory that both drugs of abuse and the consumption of highly palatable foods converge on a shared pathway within the limbic system to mediate motivated behaviors [61,62,63]. Therefore, the hedonic consumption of palatable foods involves the mesolimbic dopamine (A_10_) neurons that emanate from the ventral tegmental area (VTA) and project onto structures such as the nucleus accumbens (NAc), prefrontal cortex (PFC), hippocampus, and amygdala [64,65,66]. The VTA is comprised of three main neuronal phenotypes including dopaminergic, GABAergic, and glutamatergic neurons. The rate limiting enzyme for catecholamine synthesis is tyrosine hydroxylase (TH), which catalyzes the hydroxylation of tyrosine to 3,4-dihydroxyphenylalanine that is then rapidly decarboxylated to produce dopamine and, in some neuronal populations, norepinephrine and epinephrine [67]. Optogenetic studies have shown that local activation of A_10_ dopamine neurons or their terminals within the NAc promotes responses of increased reward-seeking behavior [68,69,70,71]. On the other hand, activation of VTA GABAergic neurons promotes aversive responses, while their inhibition promotes reward, by dampening or promoting A_10_ dopamine signaling, respectively [72,73]. Photoactivation studies, following an intracranial self-stimulation operant paradigm protocol [74], further support the role of VTA A_10_ dopamine neurons in reward-seeking. This reward-seeking role is exemplified through data showing that photoactivation of VTA glutamatergic neurons expressing cation channel rhodopsin-2 (ChR2) under a vesicular glutamate transporter-2 (VGlut2) promoter caused conditioned place preference to a photostimulation-paired chamber and motivated operant responding to earn optical intracranial self-stimulation in mice [75]. These rewarding effects are attributed to local excitatory input onto the VTA A_10_ dopamine neurons [75]. A_10_ dopamine neurons thereby encode reward processing for natural and drug-induced rewards and are implicated in increasing incentive salience for palatable foods, food-seeking behavior and impulsivity that could, under the right circumstances, lead to binge-feeding behavior [4,64,66,76,77,78].

To fully grasp the distinct role A_10_ dopamine neurons play in hedonic feeding patterns, it is imperative to delineate the multifaceted aspects of global reward processing. The influential theories proposed by Berridge and colleagues make the case that when examining the role of food reward in feeding behavior, distinctions must be made between what he coined as reward ‘liking’ and reward ‘wanting’ [79,80]. ‘Liking’ is associated with the hedonic impact, or the brain reaction underlying sensory pleasure triggered by a rewarding stimulus, such as a highly palatable food [79,80]. Hedonic pleasure (‘liking’) has been reliably measured through observation of facial affective reactions prompted by exposure to a natural taste stimulus, where sweet tastes elicit positive ‘liking’ patterns of distinct orofacial expressions (e.g., rhythmic or lateral tongue protrusions) and bitter tastes alternatively evoke “disliking’ expressions (e.g., gapes) [81]. These varying patterns of orofacial reactions are homologous to those observed in human infants, orangutan, chimpanzees, monkeys, rats, and mice, insinuating evolutionary conservation of the underlying brain circuits involved [80,81,82,83]. Data collected from taste reactivity studies allowed further insights into the neural underpinning of hedonic impact reactions, illuminating involvement of hedonic hotspots in the rostrodorsal quadrant of the medial shell of the NAc, ventral pallidum, and brainstem regions such as the parabrachial nucleus in the pons [80,84,85].

Conversely, hedonic impact (‘liking’) is distinguishable from ‘wanting’, which is related to incentive salience or motivation in reward-seeking [79,80]. Further, incentive salience (in the form of cue-triggered ‘wanting’) is mediated by separate neural networks, like those originating from A_10_ dopamine neurons in the VTA, though integration of signals from both components is mechanistically crucial to produce the full phenomenon we typically think of as reward [79,80]. As previously mentioned, VTA dopamine signaling is indicated in incentive salience, a notion vastly supported by evidence wherein rewarding stimuli leads to enhanced dopamine transmission, while suppression of dopamine signaling lessens the motivation for rewards including food, sex, and drugs [65,76,86,87,88]. However, taste reactivity studies utilizing mutant mice with genetically abolished neural dopamine, or impaired ventral striatal dopamine caused by neurochemical 6-OHDA lesions, have reiterated this point, showing no detectable effects on ability to register the pleasurability/hedonic impact of tastes in the absence of dopamine [80,89]. Recognition of these separate neuronal networks encompassing overall reward is key to be able to discern the contributing neural pathways, and perturbations of these pathways, that lead to pathological reward-seeking behaviors (e.g., in food and drug addiction). In regard our recent study described below in Section 4, we mainly focus on ‘wanting’ behaviors in hedonic feeding and modulations of A_10_ dopamine signaling in the context of food reward and binge feeding.

It should be noted that A_10_ dopamine neurons also play a critical role in aversion processing in ways that are related to their firing pattern as well as the topographical pattern of dopaminergic innervation of the NAc. In vivo photometry studies have revealed that responses to aversive stimuli are mediated by A_10_ dopamine neurons terminating in the ventromedial shell of the NAc [90]. Aversive responses are induced by excitation of these neurons that occurs, at least in part, via glutamatergic input from the lateral hypothalamus [91]. Optogenetic studies indicate that prolonged stimulation of these A_10_ dopamine neurons activates dopamine D1 and D2 receptor-bearing medium spiny neurons that, in turn, increases the firing rate of GABAergic neurons in the ventral pallidum, decreases the excitability of A_10_ dopamine neurons and reduces cocaine reward [92].

As with the hypothalamic energy balance circuit, the neurons comprising the hedonic reward circuitry, namely the A_10_ dopamine neurons, are also sensitive to circulating leptin, insulin, and ghrelin levels, relaying signals to regulate food-seeking behavior and ultimately changes in body mass [93]. Leptin binding to and activating LEPRs expressed on VTA A_10_ dopamine neurons leads to phosphorylation of STAT3, membrane hyperpolarization and reduced firing in these neurons [94,95]. Moreover, site-specific ablation of LEPRs in the VTA heightens the sensitivity of mice to the rewarding aspect of highly palatable foods (e.g., sucrose), while microinjection of leptin into the VTA reduces food intake [95]. Secondary regulation of VTA dopamine neurons by leptin is mediated via direct leptin action on LH LEPR expressing neurons which in turn relay signals onto the A_10_ dopamine population, via synaptic contact, to promote decreases in food intake and concomitant decreases in body weight [96]. Interestingly, LH LEPR neurons have been shown to represent a unique neuronal population distinct from the previously mentioned orexin or MCH neurons also found in the LH [96]. Anterograde tract- and retrograde tracing further confirmed that LH LEPR neurons project caudally to densely innervate the VTA, with few to no projections seen in hypothalamic regions (including the ARC) or in the striatum (including the NAc). Therefore, LH LEPR neurons may play a unique role in A_10_ dopamine regulation compared to other LEPR expressing neurons in the hypothalamus [96]. This interaction of LH signaling to control VTA neurons importantly highlights how homeostatic and hedonic neural circuits may dynamically and coordinately interact with one another to promote global energy balance. Insulin receptors are also present on VTA (and substantia nigra) dopamine neurons and can induce expression of the dopamine transporter (DAT) in these neuronal populations, as was demonstrated following intracerebroventricular (i.c.v.) insulin treatment [93]. Increased DAT results in quicker dopamine reuptake from the synaptic cleft back into presynaptic neurons, thereby halting stimulation of postsynaptic neurons and ultimately working to decrease the rewarding effect of food [93]. Furthermore, ghrelin binding to and activating ghrelin receptors expressed on A_10_ dopamine neurons has been shown to elevate the frequency of action potentials in these neurons, as well as induce increased dopamine turnover in the NAc, promoting appetite [97]. Interestingly, the ghrelin-induced increase in locomotor activity and dopamine release in the NAc is negated by CB1 receptor antagonism with rimonabant [98]. Thus, the abundant presence of communication between the brains reward circuitry and fluctuating hormones and neuromodulators, that typically relay nutritional/energy state cues, underscores the important role of A_10_ dopamine signaling in energy balance and potential links to feeding pathologies.

The intrinsic separation of the neural systems encoding hedonic impact from pleasure and the incentive salience or motivation for a reward, can give way to possible explanations for aberrant feeding behaviors that may underlie feeding pathologies [80]. Individuals with substance abuse disorders seem to take drugs compulsively even when they no longer derive pleasure from them (‘liking’), and their motivation to take the drug (‘wanting’) may persist due to long-lasting sensitization of their brain mesolimbic systems, brought on by repeated binges [76,80,88]. Food is a natural reward with reinforcing properties, similar to rewards such as sex and drugs, and activates the dopaminergic mesolimbic system by elevating extracellular dopamine concentrations in the NAc [99,100,101]. In particular, highly palatable, calorie rich foods can critically effect A_10_ dopamine neurotransmission [102,103], analogous to modulations caused by other potent reinforcing stimuli, e.g., cocaine, amphetamines, opiate-like drugs, cannabinoids, alcohol, and nicotine [104,105,106,107,108]. Some evidence suggests that similar sensitization-like changes can be induced by exposure to certain regimens of food and restriction, modelling oscillations between dieting and binging on palatable foods [109,110,111,112,113]. Wherein, rats exposed to brief, intermittent bouts of sucrose access (sucrose binges), express sensitization-like changes, especially when binges are cycled with food-restriction. Observed changes include: increasing propensity to over-consume when allowed, an enduring enhanced neural response to the presentation of food reward and cues, and an over-response to the psychostimulant effects of drugs such as amphetamine (a typical behavioral marker of drug-induced neural sensitization, which suggests a common underlying mechanism) [110,113,114,115]. Indeed, dopamine release into the NAc and c-fos expression in A_10_ dopamine neurons is increased during binge-feeding episodes [77,116], lending further credence to this notion. Additionally, obesity is associated with dysfunction of dopaminergic systems. Obese patients present with reduced striatal dopamine D2 receptor, as measured by positron emission tomography [117]. This similarity between food reward and drug reward gives rise to the notion that feeding disorders and drug abuse and/or dependence share common mechanisms, as neuropsychological diseases involving negative alterations of the neural networks associated with the reinforcing properties of rewards [118,119,120].

## 3. Influences of Sex and Diet on Central Energy Balance Circuits

Sex differences are abundantly present in the context of energy homeostasis and although the prevalence of obesity is similar between men and women, women seem to have a greater risk of developing eating disorders and extreme obesity [121,122]. Indeed, sex is thought to represent one of the main risk factors for food-related disorders, including binge eating disorder [123]. Other lines of evidence illustrating sex differences in food-based reward processing include women having a reduced ability to control food desire, higher cortical and limbic activation when presented with visual, gustatory and olfactory cues, as well as increased susceptibility to episodes of food-craving and lack of control for sugary foods, compared to men [124,125,126,127,128]. The role of sex hormones, particularly the fluctuating estrogen levels in females throughout the estrous cycle, is of interest in these disparities seen between males and females for reward-based consumption.

Estrogens elicit inhibitory effects on food intake, attributable to activation of estrogen receptors in key brain regions responsible for food intake control and body weight, such as the hypothalamus and nucleus tractus solitarius [5,129,130,131,132]. In regard to sex hormones, studies on the cannabinoid system have elucidated activation effects of gonadal steroid hormones on hypothalamic energy balance neural circuitry. Estradiol (E_2_) in females has been shown to attenuate cannabinoid-induced hyperphagia and hypothermia, as well the decrease in glutamatergic input onto POMC neurons [133]. These estrogenic actions occur through activation of estrogen receptor (ER) and the G_q_-coupled membrane ER (G_q_-mER), which triggers a signaling pathway involving PI3K, protein kinase C (PKC), protein kinase A (PKA) and neuronal nitric oxide synthase (nNOS) [131,132,134]. This, in turn, diminishes endocannabinoid tone at VMN SF-1/ARC POMC synapses, thereby relieving the retrograde inhibition of the glutamatergic input [10,11]. Regarding nNOS, while multiple isoforms with differing efficacies in eliciting downstream nitric oxide signaling have been characterized in brain [135], it is currently unknown which isoform mediates the estrogenic diminution in endocannabinoid tone at these synapses. Therefore, it will be necessary for future studies to elucidate the isoform responsible.

Interestingly, functional glutamatergic synapses at these VMN SF-1/ARC POMC synapses are largely silenced in obese females [11]. Indeed, studies of both POMC and NPY/AgRP neurons from *ob/ob* mice reveal that they undergo extensive synaptic plasticity under conditions of obesity; with the former receiving significantly more inhibitory inputs concomitant with a reduction in excitatory inputs, and the latter having appreciably more excitatory inputs and fewer inhibitory ones impinging upon them [136]. Obesity also correlates with chronic inflammation and resistance to leptin and insulin not only in the CNS but also peripherally [137,138,139,140]. However, the inflammation, reactive gliosis and subsequent neuronal injury observed in the mediobasal hypothalamus develops more rapidly than for similar maladaptations occurring in peripheral organs [141].

By contrast, testosterone in males rapidly increases energy intake and is reversed by the CB1 receptor antagonist AM251 and the diacylglycerol lipase (DAGL) inhibitor Orlistat, and potentiates the cannabinoid-induced decrease in glutamatergic input onto POMC neurons [10,142]. This androgen-induced increase in endocannabinoid tone is due to activation of AMPK, which augments retrograde inhibition of glutamate release at VMN SF-1/ARC POMC synapses [10,142]. These effects are further magnified in obese males due to reduced PI3K signaling in the ARC [11,143]. Lastly, the development of central insulin resistance brought on by diet-induced obesity is sexually differentiated. Males are more susceptible to the attenuated activation of both TRPC5 channels in POMC neurons and K_ATP_ channels in NPY/AgRP neurons than are females under the protection of E_2_, which prevents the respective increase in suppressor of cytokine signaling-3 and decrease in PI3K signaling that drives central insulin resistance [143,144].

Estrogen receptors are also expressed within the VTA. Evidence indicates that estrogens increase self-administration of rewards like psychomotor stimulants and alcohol [145,146,147]. In animal studies, female rats have been more motivated to work for cocaine during the estrus phase, compared to other phases of the estrous cycle [148,149], and E_2_-treated ovariectomized (OVX) female rats exhibited increased motivation to self-administer cocaine [150]. The effects of estrogen on reward neural circuitry are also evident in motivation for food rewards. In opposition of the findings from self-administration of drug rewards, intra-VTA injections of E_2_ significantly reduced the motivation to work for sucrose rewards in a progressive ratio operant conditioning task within 1 h after injection, while overall food intake was not altered by this treatment [151]. Additionally, a study comparing the self-administration of chocolate-flavored beverage (CFB) and concomitant changes in extracellular dopamine in the dialysate obtained from the NAc, between male as well as intact and OVX female rats, showed that female rats in the proestrus and estrus phases of the cycle had reduced lever responding for, and amount of self-administered CFB, paired with lowered extracellular dopamine in the dialysate from the NAc shell [152]. These variable findings between food versus drug rewards raise questions about the role estrogens play in food reward processing and how they may potentially explain the disparate prevalence rates between males and females in feeding behavior pathologies.

## 4. Nociceptin/Orphanin FQ Regulation of Homeostatic and Hedonic Energy Balance Circuits

The neuropeptide nociceptin/Orphanin FQ (N/OFQ) is an endogenous opioid hepadecapeptide that is encoded by the prepronociceptin gene [153,154,155]. N/OFQ binds with high affinity to its cognate G_i/o_-coupled nociceptin opioid peptide (NOP) receptor, and despite high structure homology, has minimal affinity for classic opioid receptors (mu, kappa, or delta opioid receptors), nor do classical opioid receptor ligands (e.g., naloxone, endorphin, dynorphin) have high affinity binding for NOP [155,156]. Initial studies on N/OFQ indicated the peptide attenuates locomotor activity, increases sensitivity to pain, while blocking the antinociceptive activity of mu, delta, and kappa analgesics following i.c.v. injections; thus garnering the name nociceptin [154,155,157]. The NOP receptor is expressed extensively throughout the central nervous system (CNS) [153,156,158,159], subserving a role in an array of central processes including pain, learning and memory, emotional states, neuroendocrine control, food intake, and motor control (see [160] for thorough review).

The actions of N/OFQ in these disparate neuronal systems is accredited to peptide binding and subsequent activation of its NOP receptor. In line with agonist activation in all GPCRs, following NOP activation by N/OFQ the Gα and Gβγ subunits dissociate to then promote the various effector pathways [155,158]. Therein, NOP receptor activation inhibits adenylate cyclase (AC) activity and couples to pertussis-toxin-sensitive G-proteins resulting in decreased cyclic adenosine monophosphate (cAMP) production [154,155,160,161]. Further, NOP receptors canonically couple to G protein-gated inwardly rectifying potassium (GIRK)3 and both N-type as well as P/Q type voltage-gated Ca2+ channels. Upon NOP receptor activation, Ca2+ currents are reduced and GIRK channels activated, causing K^+^ efflux, cellular hyperpolarization and attenuated neural activity [162,163,164]. Additionally, NOP receptor signaling has been shown to promote activation of PKC as well as phospholipase A2 and C [165,166,167].

As mentioned, the N/OFQ/NOP receptor system is expressed widely throughout the brain in mice, rats, and humans, and importantly is expressed densely in the ARC and VTA [153,156,159,168,169]. The abundant expression of N/OFQ and NOP in these key regions supports a role in both homeostatic and hedonic energy balance regulation. Indeed, i.c.v. injections into the lateral ventricle (LV), 3V, and intranuclear injections into the ARC of N/OFQ or NOP agonists have been shown to produce a dose-dependent increase in feeding behavior even in satiated animals; intra-ARC injections proving to be the most efficacious to induce hyperphagia [170,171,172,173,174]. Additionally, chronic i.c.v. infusions of the neuropeptide have been associated with increased body weight via modifications in feeding and metabolism in mice [175]. In addition, we have found that the rebound hyperphagia seen upon refeeding in fasted NOP receptor null mice is significantly blunted compared to their wildtype littermate controls [176]. The appetite-stimulating properties of N/OFQ can be attributed, at least in part, to the NOP receptor-mediated inhibition of POMC neurons via activation of GIRK channels, as seen with either exogenous bath application of the neuropeptide or high-frequency optogenetic stimulation of ARC N/OFQ neurons [5,176,177,178,179]. The observed effects of N/OFQ are not limited to anorexigenic pathways, however, and have been observed to also influence orexigenic circuitry. NOP receptors are expressed in NPY/AgRP neurons, and N/OFQ increases AgRP release from mediobasal hypothalamic explants [180]. Interestingly, these ARC N/OFQ neurons co-release GABA upon low-frequency optogenetic stimulation, which can provide additional inhibitory input onto POMC neurons [181]. They are also regulated by ambient levels of extracellular glucose, and inhibited by leptin [181]. On the other hand, N/OFQ administered into the perifornical/lateral hypothalamic area exerts a hypophagic effect attributed to a NOP receptor-mediated inhibition of orexin neurons via activation of K_ATP_ channels [182]. Nevertheless, the prevailing sentiment based on the totality of the available evidence clearly indicates that N/OFQ is profoundly hyperphagic via its collective actions within the homeostatic energy balance circuitry.

Moreover, several lines of in vivo and in vitro evidence illustrate the effects of N/OFQ on the A_10_ dopamine system, providing credence to the notion that N/OFQ may be inherently linked to reward system processing, and thus to hedonic feeding behavior. Data obtained from in vivo studies have shown that N/OFQ decreases the outflow of dopamine in the NAc following intracerebral injections, dampens the morphine-induced dopamine release in the NAc, and blocks the acquisition of morphine-dependent place preference [183,184,185,186]. Further, bath application of N/OFQ during intracellular electrophysiology recordings in rat midbrain slices caused membrane hyperpolarization and reduced firing under current-clamp, which was associated with an outward current under voltage clamp [187]. These inhibitory effects of N/OFQ in the VTA were reduced by the NOP receptor antagonist [Phe11jCH2-NH)Gly2]NC(1 ± 13)NH2 (1 μM) but were unaltered by presence of tetrodotoxin or the opioid receptor antagonist naloxone [187]. Additionally, activation of NOP receptors expressed in the NAc and dorsal striatum work to pre-synaptically inhibit dopamine synthesis and tyrosine hydroxylase (TH, the rate-limiting enzyme in dopamine production) phosphorylation and act post-synaptically on dopamine responsive neurons by decreasing dopamine D1 receptor signaling via suppression of cAMP/PKA activity [188]. Lastly, microdialysis studies have revealed that N/OFQ significantly reduces extracellular DA levels in both the VTA and NAc [184].

The preceding section on sex differences in energy balance control is particularly relevant when it comes to NOP receptor-mediated regulation. In homeostatic energy balance, males are significantly more sensitive to the inhibitory effects of exogenous N/OFQ on excitatory neurotransmission at VMN SF-1/ARC POMC synapses than are females during E_2_-dominated phases of the estrous cycle in two ways: (1) the direct hyperpolarization/cessation of firing of both VMN SF-1 and ARC POMC neurons, as well as the underlying outward current, due to the activation of GIRK channels is greater in males than in females and (2) the presynaptic inhibition of glutamate released of VMN SF-1 neurons at these synapses is more substantive in males than in females [5,189,190]. Additionally, E_2_ exerts powerful activational effects by diminishing the inhibitory effects of exogenous N/OFQ at VMN SF-1/ARC POMC synapses, as well as the decreased excitability of POMC neurons caused by optogenetic stimulation of ARC N/OFQ neurons, and protecting against the aberrant hyperphagia and reduction in energy expenditure caused by exogenous N/OFQ administered directly into the ARC of obese OVX females [5,178,179,191]. E_2_ also attenuates the pleiotropic actions of N/OFQ on POMC neurons by binding to either ERα or the G_q_-mER, which leads to a signaling cascade that includes phospholipase C (PLC), PI3K, PKC, PKA and nNOS [191]. Furthermore, progesterone administered to OVX, estrogen-primed females restores the sensitivity of POMC neurons to these pre- and postsynaptic actions of N/OFQ [192]. The precise actions of N/OFQ and NOP receptor activation in the aforementioned neural circuitry, its effects on feeding-behavior, and potential role in feeding-related pathologies will be discussed in further detail below.

In addition to sex differences, diet modifications resulting lean or obese phenotypes alter the signaling effects of N/OFQ in homeostatic and hedonic neural circuits, and increases the risk for aberrant feeding-behavioral patterns to form. Diet-induced obesity increases the intrinsic excitability of ARC N/OFQ neurons, which augments the inhibitory GABAergic tone received by POMC neurons [181]. Conversely, ablation of ARC N/OFQ neurons hinders the development of obesity caused by a four-week exposure to a HFD [181]. Moreover, diet-induced obesity renders VMN SF-1/ARC POMC synapses more susceptible to the inhibitory effects of exogenous N/OFQ in males and hypoestrogenic OVX females [5]. This greatly curtails anorexigenic signaling at these synapses; causing exaggerated N/OFQ-induced increases in energy intake and decreases in energy expenditure, and is entirely in keeping with our recent demonstration that N/OFQ administered directly into the ARC significantly enhances binge-feeding behavior caused by short-term intermittent exposure to a HFD [179].

Endogenous N/OFQ signaling may also be intrinsically associated with or escalate aberrant feeding patterns associated with diet-induced obesity and/or binge-feeding. In one instance, Sprague Dawley rats that had previously been determined as “fat-preferring”, were particularly susceptible to N/OFQ-induced acute hyperphagia following i.c.v. injection [173]. Additionally, NOP receptor knockout mice displayed significantly reduced levels of HFD food consumption, compared to their wildtype littermate controls [193]. Additionally, administration of the novel NOP antagonist LY2940094, effectively increases lipid utilization metabolism and reduces fasting-induced hyperphagia of chow in wildtype 129S6 mice, but not in those with genetic deletion of the NOP receptor [194,195]. Further, LY2940094 reduces HFD consumption measured over a 5-h exposure period and also hindered weight gain over 3 days of HFD exposure [195]. Providing further support for the intrinsic contribution of endogenous N/OFQ/NOP signaling in the pathogenesis of obesity and eating disorders, systemic administration of LY2940094 reduced intake of HFD in diet-induced obese rats and mice, and also improved metabolic parameters by reducing the respiratory quotient in mice with access to HFD in their metabolic feeding chamber [195]. In relation to N/OFQ and binge-feeding, systemic treatment with the selective NOP antagonist SB 612111 produced a dose-dependent decrease in intermittent HFD binge eating, but not a change in the total 24-h food intake of mice who were either on an intermittent-HFD or continuous-HFD feeding regimen [196].

While the last few studies mentioned regarding altered feeding behavior certainly provide evidence that the N/OFQ-NOP system is involved in the neuropathology of obesity and related eating disorders, they were not designed to offer insight into the specific neuronal populations acted upon that leads to such pathogeneses. In addition, the exact mechanisms through which sex and diet interact to modulate NOP receptor-mediated inhibition of reward encoding A_10_ dopamine neurons and hedonic feeding remained a mystery until very recently. We have discovered that the endogenous release of N/OFQ caused by high-frequency optogenetic stimulation of VTA neurons in mesencephalic slices from N/OFQ-cre mice powerfully inhibits neighboring VTA neurons; an effect that is faithfully recapitulated by exogenous bath application of N/OFQ during recordings of A_10_ dopamine neurons in slices from TH-cre mice [179]. The membrane hyperpolarizations and underlying outward currents were attenuated by E_2_ in females, and accentuated by diet-induced obesity in males [179]. These N/OFQ-induced inhibitory effects on A_10_ dopamine neurons functionally translated into sex- and diet-dependent changes in binge-eating behavior, as N/OFQ delivered into the VTA decreased the rampant consumption seen during the binge episodes in obese but not lean males, and in both lean and obese females [179]. Just as we saw with the inhibitory effects of N/OFQ in A_10_ dopamine neurons from female mice, E_2_ counteracted the inhibitory effect of intra-VTA N/OFQ on binge feeding [179]. Thus, it is clear that N/OFQ exerts multifaceted effects on energy balance via NOP receptor-mediated regulation of homeostatic and hedonic circuits that are site-specific as well as sex hormone- and diet-dependent. The site specificity underscores the fact that despite the inhibitory effect of N/OFQ on A_10_ dopamine neurons and the associated dampening of binge-feeding behavior, there is clearly a net hyperphagic response caused by the peptide that is due largely, if not exclusively, to its effects within the homeostatic energy balance circuitry.

## 5. Pituitary Adenylate Cyclase-Activating Polypeptide (PACAP) Regulation of Homeostatic and Hedonic Energy Balance Circuits

Pituitary adenylate cyclase-activating polypeptide (PACAP) is a hypophysiotropic neurohormone belonging to the largest family of developmental and regulatory peptides that includes vasoactive intestinal polypeptide (VIP)-secretin-growth hormone-releasing hormone (GHRH)-glucagon superfamily, along with secretin, glucagon, human growth hormone-releasing factor (hGRF), and VIP [197]. Containing the most conserved sequence throughout evolution in its family, PACAP is encoded by the ADCYAP1 gene as a 38-amino acid C-terminally ⍺-amidated polypeptide [197]. It is transcribed in hypothalamic neurons and peripheral organs, such as the GI tract, pancreas, and gonads; exerting pleiotropic physiological effects such as the regulation of neurotransmitter release and secretion, vasodilation, energy balance, as well as stimulation and inhibition of cell proliferation and/or differentiation [198,199,200]. The PACAP precursor yields two different forms of PACAP, PACAP_1–38_ and PACAP_1–27_, as well as PACAP-related peptide (PRP) [200]. Given its widespread distribution in the CNS and periphery, PACAP is well-equipped to act as a hormone, a neurotransmitter, and a trophic factor in various tissue types [200].

The effects of PACAP are mediated through two classes of PACAP receptors: the pituitary adenylate cyclase-activating polypeptide-specific 1 receptor (PAC1) and two subtypes of VIP/PACAP-receptors termed VPAC1 and VPAC2 [197,200]. PAC1 receptor exhibits almost a twofold higher affinity for PACAP than for VIP, while the VPAC1 and VPAC2 receptors recognize both PACAP and VIP with equally high affinity [197]. PAC1, a metabotropic receptor, is found in various hypothalamic structures including the supraoptic nucleus (SON), PVN, ARC, LH, VMN, as well as extrahypothalamic regions of the brain like the cerebral cortex, Broca’s area, the hippocampus, among others [197,200]. VPAC1 and VPAC2 receptors are appreciably expressed in peripheral organs including the lung, duodenum, and thymus, although with less abundance than PAC1 receptors [201,202]. There is considerable evidence demonstrating that the PAC1 receptor signals through G_q_ and G_s_. For example, physiological studies have determined that PACAP acts on PAC1 receptor in mouse neural stem cells, and upon receptor activation the signal generated is carried via a G_q_-mediated PLC/diacylglycerol/inositol 1,4,5-trisphosphate (IP3)-dependent signaling pathway [203]. In the hypothalamus, PACAP binds to PAC1 receptors that induce G_q_-coupled stimulation of PLC, PI3K, and PKC to ultimately activate TRPC5 channels [12]. In the neurohypophysis, activation of PAC1 receptors by PACAP can lead to signaling via the G_s_ pathway [204] that increases firing and depolarizes the membrane potential of magnocellular neurons in rat brain slices via the activation of adenylate cyclase/cAMP/PKA signaling [204,205,206]. In doing so, PACAP stimulates the release of oxytocin and vasopressin from the posterior pituitary [204,207]. Activation of PAC1 receptors has also been shown to elevate intracellular Ca^2+^ concentrations in dissociated magnocellular neurons from rat SON, and stimulate a quinine- sensitive K^+^ outward current in murine microglia [207,208].

In the immune system, PACAP decreases chemotaxis of thymocytes and lymphocytes both via the activation of the PKA pathway [209,210]. Additionally, PACAP inhibits tumor necrosis factor-α and both interleukin-6 and interleukin-12 release, while enhancing the production of the cytokine interleukin-10 in lipopolysaccharide-activated macrophages; suggesting it acts as a protective agent that regulates the release of proinflammatory and anti-inflammatory cytokines [211,212,213,214]. The effects of PACAP on cell proliferation/survival are dependent on the downstream channel targeted as well as transcriptional cues. In vitro experiments have demonstrated that the effect of PACAP on cell survival is regulated via the activation of the G_s_ pathway, contributing to the phosphorylation of the extracellular signal-regulated (ERK)-type of mitogen-activated protein (MAP) kinase and enhanced c-fos gene expression [215,216,217]. On the other hand, PACAP dose-dependently inhibits concanavalin A-induced cell proliferation in murine splenocytes [218]. Moreover, PACAP stimulates Ca^2+^ mobilization and blocks K^+^ currents in a variety of neuronal cell types (e.g., magnocellular neurons, cerebellar granule cells), two processes intimately linked to PACAP-induced enhancements in cell excitability [206,219,220,221]. Furthermore, PACAP stimulates postprandial glucagon-like peptide, leptin and insulin secretion, and also has been shown to promote insulin secretion from pancreatic β-cells via Ca^2+^ influx through L-type Ca^2+^ channels [222,223,224,225,226]. This latter effect may be dependent on ambient glucose concentration and its ability to enhance ATP production and thereby negatively gate ATP-sensitive potassium (K^+^_ATP_) channels [227,228]. This indicates that endogenous PACAP acts as a physiological regulator of pancreatic β-cell activity linked to K^+^_ATP_ channels in a manner similar to that described for the vasodilatory and neuroprotective effects of the peptide. [229,230]. Thus, it is clear that PACAP regulates a wide array of bodily functions including hypophysiotropic neurosecretion, glial function, immunomodulation, cell proliferation/survival, glucose homeostasis, vasodilation, neuroprotection and energy homeostasis via G_s_- and G_q_-mediated signaling. Moreover, given the inexorable link between obesity and inflammation [137,138,139,141], it is entirely conceivable that the overall anti-inflammatory effect of PACAP contributes significantly to lean phenotypes promoted by this peptide [226,231].

PACAP exerts myriad effects on energy balance at all levels of the brain-gut axis. For example, intravenous injection of PACAP on rats causes secretion of saliva from the submandibular and parotid glands, whereas in the stomach, PACAP decreases histamine- and pentagastrin-activated gastric acid secretion; the latter suggesting that PACAP acts indirectly to regulate gastric acid release [232,233,234]. In addition, intravenous injection of PACAP increases bicarbonate secretion and chloride secretion in the duodenum and in the distal colon, respectively [235,236]. Moreover, while PACAP administered centrally increases gastric motility, peripherally it evokes a dose-dependent relaxation of the gastric smooth muscles, decreases gastric motility and therefore delays stomach clearing [237,238,239]. Likewise, PACAP stimulates intestinal smooth muscles to relax in rats and other species [240,241,242,243].

Concerning peripheral glucose and lipid homeostasis, PACAP exerts a more potent action in stimulating glucose output from a perfused rat liver as compared to VIP [244]. While PACAP can clearly act centrally to regulate glucose homeostasis, the hyperglycemic role of PACAP in vivo can also be attributed to both an indirect action via increase in plasma glucagon and/or catecholamines, which increase glycogenolysis and gluconeogenesis [245]. Finally, in regard to energy metabolism, PACAP is known to accelerate lipolysis via the sympathetic nervous system (SNS). This suggests that hypothalamic PACAP signaling may promote the use of catabolized lipids as a viable energy source [197,231].

The hypothalamic regions that play a role in the regulation of thermogenesis, energy expenditure, and energy intake such as the PVN, VMN, and ARC abundantly express PACAP and the PAC1 receptor, suggesting that PACAP plays a vital role in the control of these processes [200,201,246]. There are two major populations of hypothalamic PACAP neurons—one with cell bodies residing in the PVN and the other with somata in the VMN [9,12,247]. The PVN PACAP neurons are thought to promote appetite through synaptic connections with, and excitation of, NPY/AgRP neurons [9]. It is the loss of signaling via this population of PACAP neurons that may explain reports of hypoinsulinemia, decreased adiposity, lower body weight and increased insulin sensitivity seen in transgenic PACAP-null mice [248]. The VMN PACAP neurons exhibit extensive colocalization with SF-1, which along with leptin drives PACAP expression [12,247,249]. These neurons are reported to be glucose inhibited, and their selective activation reduces circulating insulin concentrations and glucose tolerance [250,251]. VMN PACAP neurons make synaptic contact with POMC neurons, and like POMC neurons, are excited by leptin [9,12,36,37,252]. As such, this VMN population of PACAP neurons is poised to suppress appetite and enhance energy expenditure. In accordance, studies show that PACAP injected into the VMN causes an increase in body temperature via adaptive thermogenesis and increased levels of UCP-1 [197,247,253,254]. The adaptive thermogenesis brought on in part by increasing WAT browning in rodents increases energy expenditure and suppresses diet-induced obesity and glucose intolerance [49,50]. PACAP also controls activity, as an injection of PACAP given i.c.v. or into the VMN increases locomotion in rodents concomitant with an increase in O_2_ consumption [247,253,254]. A systemic injection of PACAP into wild type (WT) mice dose-dependently lowers cumulative energy intake and decreases various indices of meal pattern like meal-size and rate of consumption, which correlates with reduced ghrelin secretion [226]. PACAP delivered directly into the VMN or PVN also reduces energy intake; however, these effects are coupled with somewhat disparate effects on meal pattern. PACAP injected directly into the PVN decreases meal size, rate of consumption, duration, total time spent eating and increased latency to meal initiation, whereas PACAP injected into the VMN only evokes an increase in the latency to meal initiation and a decrease in the rate of eating [254]. Thus, PACAP signaling throughout the homeostatic energy balance circuitry exerts far-reaching effects on energy intake, meal pattern and energy expenditure.

Additional studies investigating the homeostatic energy balance circuitry have demonstrated that PAC1 receptors are expressed in POMC neurons, and that PACAP administered to ad libitum-fed animals elevates POMC expression, c-Fos expression in POMC neurons, and MC4R receptor mRNA expression when injected i.c.v. or into the VMN, as well as enhancing α-MSH release from hypothalamic explants [250,253,255]. In contrast, PACAP had no effect on AgRP, CART or NPY mRNA levels [250]. In addition, PAC1 receptor blockade with PACAP_6–38_ or PACAP deficiency seen in Adcyap1^-/-^mice significantly decreases the leptin-induced hypophagia, hyperthermia, and increase in WAT sympathetic nerve activity in vivo [231,247]. We have recently shown that the profound influences PACAP exerts in the homeostatic control of energy balance in ad libitum-fed mice are diet- and sex-dependent [12]. We observed that PACAP evokes an inward current associated with an increase in firing in ARC POMC neurons that was abolished by PAC1 receptor antagonism and TRPC5 channel blockade, and augmented by E_2_ [12]. The inward current was significantly attenuated upon inhibition of PLC, PI3K and PKC, but not PKA; suggesting that the PACAP-induced activation of POMC neurons was mediated via, PI3K and G_q_-mediated signaling pathways [12]. The stimulation of ARC POMC neurons caused by PACAP administered directly into the ARC suppresses energy intake and enhances energy metabolism, and these effects were markedly attenuated under conditions of diet-induced obesity [12]. These effects of exogenously applied PACAP were effectively mirrored by chemogenetic and optogenetic stimulation of VMN PACAP neurons [12]. Collectively, these data suggest that under ad libitum-fed conditions PACAP functions through a PI3K/PLC/PKC pathway to activate POMC neurons via TRPC5 channels upon PAC1 receptor stimulation via G_q_-mediated signaling [12]. Thus, VMN PACAP/ARC POMC synapses constitute a critical anorexigenic component of the homeostatic energy balance circuitry, one that is accentuated by E_2_ in females and attenuated by obesity in males. These findings are consistent with other examples of positive estrogenic modulation of the PACAP/PAC1 receptor system that occur during the response to stress. E_2_ stimulates PACAP and PAC1 receptor expression in the bed nucleus of the stria terminalis (BTNS) as well as in the medial basal hypothalamus (MBH) compared to levels seen in the oil-treated OVX controls [256]. Likewise, PACAP expression in the PVN and anterior pituitary is heightened during the elevated E_2_ levels seen during proestrus [256]. Concerning the hedonic energy balance circuitry and the consumption of palatable food, prior studies have shown that when PACAP is injected into the NAc it mimics the inhibitory effect of GABA receptor agonists on binge-like feeding behavior and decreases firing in NAc neurons [257]. This effectively reduces hedonic drive for palatable food as gauged by decreases in appetitive orofacial responses to sucrose, as well as by increases in aversive responses when PACAP is administered into the caudal NAc [258].

Whereas obesity exemplifies a state of positive energy balance, fasting represents a state of negative energy balance. Food-restricted mice show low levels of POMC and PACAP mRNA expression coupled with an increase in NPY mRNA expression, and i.c.v. injections of PACAP decrease energy intake after 30 min of refeeding [247,250]. Surprisingly, in recent experiments we found that during voltage clamp recordings in POMC neurons from eGFP-POMC mice subjected to an 18-h fast for five consecutive days, the PACAP response reversed polarity from a predominantly excitatory inward current seen under ad libitum conditions to an inhibitory outward one (Figure 2A,B,E–H); see Appendix A for full material and methods The inset I/V plots corresponding to the representative current traces in Figure 2A,B revealed that the reversal potential shifted from ~−20 mV under ad libitum conditions (indicative of a mixed cation current) to ~−90 mV under fasted conditions (indicative of a K^+^ current). The PACAP-induced inhibitory outward current and the corresponding increase in conductance seen under fasting conditions were virtually abolished by the K_ATP_ channel blocker tolbutamide and the PAC1 receptor antagonist PACAP_6–38_ (Figure 2C,D,G,H; 2G: one-way ANOVA/LSD, F = 25.71, df = 3, *p* < 0.0001; 2H: one-way ANOVA/LSD, F = 9.78, df = 3, *p* < 0.0001).

A nearly identical switch in polarity of the PACAP response in POMC neurons was observed during voltage clamp recordings in EtOH vehicle-treated slices from fasted OVX female eGFP-POMC mice (Figure 3A,C–E). Bath application of E_2_ in slices from fasted OVX female eGFP-POMC mice reduced the magnitude of the PACAP-induced outward current by ~50%, and markedly attenuated the increase in K^+^ conductance (Figure 3B,D: Student’s *t*-test, *t* = 1.51, *p* < 0.15; 3E: Student’s *t*-test, *t* = 2.664, *p* < 0.02).

Current clamp recordings in POMC neurons from fasted male eGFP-POMC mice uncovered a more hyperpolarized resting membrane potential (RMP) than those from their ad libitum-fed counterparts (Figure 4A: Student’s *t*-test, *t* = 2.237, *p* < 0.04). Once again, the switch in polarity of the PACAP response was evident, such that the PACAP-induced depolarization of POMC neurons and the associated increase in firing seen in the representative trace and composite data from the ad libitum-fed state transformed into a hyperpolarization and a suppression of firing in the fasted state (Figure 4B–E; 4D: Student’s *t*-test, *t* = 9.501, *p* < 0.0001; 4E: Kruskal–Wallis/median-notched box-and-whisker analysis, test statistic = 15.6201, *p* < 0.002).

We corroborated these findings during optogenetic recordings in POMC neurons (Figure 5A,C,D) from PACAP-cre/eGFP POMC mice. Selective activation of VMN PACAP neurons (Figure 5B,E) elicited inward currents that were associated with depolarizations and increases in firing in the ad libitum-fed state (Figure 5F,H,J,L–N), and outward currents that were associated with hyperpolarizations and cessation of firing in the fasting state (Figure 5G,I,K–N; 5L: Student’s *t*-test, *t* = 6.444, *p* < 0.0001; 5M: Student’s *t*-test, *t* = 6.97, *p* < 0.0001; 5N: Kruskal–Wallis/median-notched box-and-whisker analysis, test statistic = 13.8222, *p* < 0.004).

Given that the tyrosine phosphatases PTP1B and TCPTP as well as AMPK figure prominently in orchestrating the cellular response to a state of negative energy balance, we then evaluated the role these signaling molecules play in the reversed polarity of the PACAP response in POMC neurons seen during fasting. In slices from fasted male eGFP-POMC mice we found that pretreatment with the PTP1B/TCPTP inhibitor CX08005 dramatically switched the PACAP response in POMC neurons from robust and reversible outward currents and hyperpolarizations (Figure 6A,C,E–I) to robust and reversible inward currents and depolarizations typically seen in the ad libitum-fed state (Figure 6B,D–I; 6E: Student’s *t*-test, *t* = 8.194, *p* < 0.0001; 6F: Student’s *t*-test, *t* = 6.878, *p* < 0.0001; 6G: Kruskal–Wallis/median-notched box-and-whisker analysis, test statistic = 13.5957, *p* < 0.004).

An equally dramatic switch in the polarity of the PACAP response in POMC neurons was observed during voltage clamp recordings in slices from both fasted eGFP-POMC mice that were pre-treated with AMPK inhibitor Compound C (Figure 7A,B,D–F), and from ad libitum-fed animals that were pre-treated with the AMPK activator metformin (Figure 7C,D,G; 7D: one-way ANOVA/LSD, F = 41.35, df = 2, *p* < 0.0001).

Consistent with these changes in cellular signaling, we also observed through immunohistofluorescent staining of coronal hypothalamic slices that fasting significantly increased the number of PTP1B- and pAMPK-positive ARC neurons as compared to that seen in the ad libitum-fed state (Figure 8; 8C: Student’s *t*-test, *t* = 7.748, *p* < 0.002; 8F: Student’s *t*-test, *t* = 5.598, *p* < 0.006).

Moreover, the flipped PACAP-induced change in the excitability of POMC neurons yielded parallel alterations in energy intake. Indeed, the pronounced anorexigenic response generated by PACAP administered directly into the ARC of ad libitum-fed wildtype male mice (Figure 9A; repeated measures multifactorial ANOVA/LSD, F_time_ = 45.21, df = 1, *p* < 0.0001; F_PACACP_ = 71.82, df = 1, *p* < 0.0001; F_interaction_ = 0.72, df = 1, *p* < 0.40) was completely reversed in fasted animals, such that there was no change in cumulative energy intake at three hours post-injection and a significant increase six hours after administration (Figure 9B; repeated measures multifactorial ANOVA/LSD, F_time_ = 28.18, df = 1, *p* < 0.0001; F_PACAP_ = 4.32, df = 1, *p* < 0.04; F_interaction_ = 4.79, df = 1, *p* < 0.04; one-way ANOVA/LSD, F = 12.88, df = 3, *p* < 0.0001). A similar profile was observed in OVX wildtype female mice; with PACAP decreasing cumulative energy intake in ad libitum-fed, sesame oil vehicle-treated OVX females, which was significantly potentiated in estradiol benzoate (EB)-treated OVX females (Figure 9C; repeated measures multi-factorial ANOVA/LSD: F_time_ = 417.17, df = 1, *p* < 0.0001), F_PACAP_ = 9.00, df = 1, *p* < 0.004, F_EB_ = 20.59, df = 1, *p* < 0.0001, F_interaction_ = 0.91, df = 1, *p* < 0.35). Conversely, the decrease in cumulative energy intake caused by PACAP in ad libitum-fed, vehicle-treated OVX females was once again completely reversed under fasting conditions. Surprisingly, EB per se failed to exert its prototypical anorexigenic effect in fasted OVX females, but it did abrogate the PACAP-induced increase in cumulative energy intake seen at six hours post-administration (Figure 9D; repeated measures multi-factorial ANOVA/LSD: F_time_ = 74.67, df = 1, *p* < 0.0001), F_PACAP_ = 8.64, df = 1, *p* < 0.004, F_EB_ = 3.18, df = 1, *p* < 0.08, F_interaction_ = 14.88, df = 1, *p* < 0.0003; one-way ANOVA/LSD, F = 15.53, df = 7, *p* < 0.0001).

Taken together, this demonstrates that under conditions of negative energy balance, PAC1 receptor/effector coupling reverts from TRPC5 channel-induced excitation to K_ATP_ channel-induced inhibition, which completely reverses the effect of PACAP on energy intake. Thus, it is clear that postsynaptic PAC1 receptors at VMN PACAP/ARC POMC synapses effectively serve as metabolic switches that provide flexibility in the face of dynamic changes in energy status.

## 6. Concluding Remarks

To summarize, the N/OFQ/NOP and PACAP/PAC1 systems exert pleiotropic actions in the homeostatic and hedonic regulation of energy balance. In short, the N/OFQ/NOP system elicits a net orexigenic effect via the homeostatic energy balance circuitry, and dampens the consumption of palatable food via inhibitory actions within the hedonic energy balance circuitry. These effects are sexually differentiated, accentuated by diet-induced obesity in males and hypoestrogenic females, and attenuated by E2 in OVX females. On the other hand, the PACAP/PAC1 system contributes a net anorexigenic effect via the homeostatic and hedonic energy balance circuitries. A thorough investigation of the anorexigenic VMN PACAP/ARC POMC synapse reveals that the neurophysiological and accompanying behavioral effects described above are diminished by diet-induced obesity in males, potentiated by E2 in OVX females, and completely reversed under fasting conditions. The manner in which alterations in energy balance status can influence NOP and PAC1 receptor-mediated signaling in POMC neurons is summarized schematically in Figure 10.

Over the past few decades, the neuroscience and neuroendocrine communities have made great strides in advancing our understanding of how the brain coordinates energy balance regulation via the homeostatic and hedonic energy balance circuitries. With the plethora of methodological tools currently at our disposal (e.g., optogenetics, chemogenetics, proteomics and transcriptomics, to name but a few), it is more than reasonable to expect that further advances will be readily and rapidly achieved. Just like the case made for PACAP in this very piece, it will be imperative to systematically evaluate all of the major players implicated in regulating energy homeostasis (including, but certainly not limited to, endocannabinoids, N/OFQ, NPY/AgRP neurons, POMC neurons, A10 dopamine neurons) not only under normophysiologic conditions but also under negative (e.g., fasting) and positive (e.g., obesity) energy balance states. Only in this way will we develop a comprehensive picture of how all of these functioning components are altered under these adaptive (e.g., fasting) and maladaptive (e.g., obesity) scenarios. In addition, it will be equally important to thoroughly dissect the impact of sex on the functioning of the central energy balance circuitries, and the cross-talk between gonadal hormones, the neurotransmitter/modulator systems and the signaling molecules involved. One final critical crowning achievement will be realized once we gain a firmer grasp on how the output from the homeostatic and hedonic energy balance circuitries functionally translates into clearly defined changes in the brain-gut axis and autonomic tone, as well as changes in gustatory and motivated behavior. We currently stand on the shoulders of pioneering giants who have paved the way by elucidating the mechanisms that provide the basis of our current understanding involved in the central control of energy homeostasis. In looking to the future, we welcome the next generation of innovative scientists to carry the torch and further advance our understanding.

## Figures and Tables

**Figure 1 ijms-22-02728-f001:**
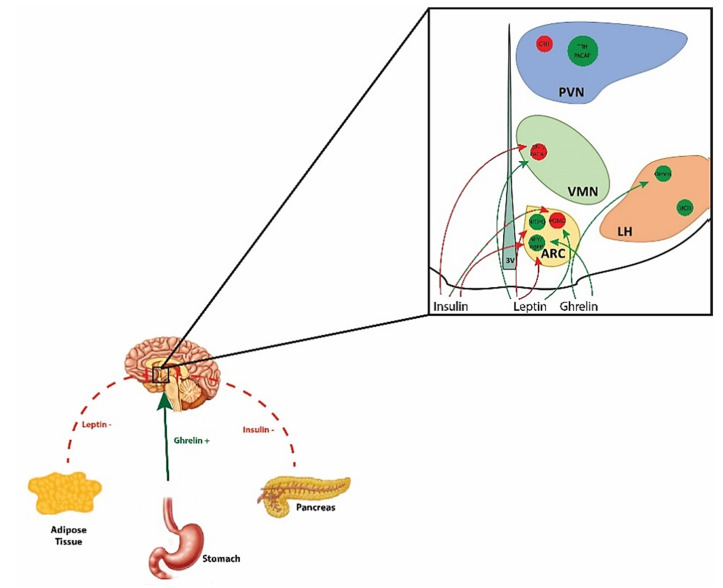
Schematic diagram illustrating the interplay between peripheral hormones and the homeostatic energy balance circuitry. Hormones like leptin and insulin released from adipose tissue and the endocrine pancreas, respectively, exert anorexigenic effects, whereas ghrelin released from the gastric mucosa exerts orexigenic effects. Leptin’s appetite-suppressing actions are mediated via excitatory effects on anorexigenic POMC and SF-1/PACAP neurons, as well as inhibitory effects on orexigenic NPY/AgRP and N/OFQ neurons. Insulin also inhibits NPY/AgRP neurons and, paradoxically, SF-1/PACAP neurons as well. Insulin’s effects on POMC neurons are dependent upon prevailing levels of tyrosine protein phosphatases. On the other hand, ghrelin’s appetite-promoting effects are due to its excitatory effects on NPY/AgRP and orexin neurons.

**Figure 2 ijms-22-02728-f002:**
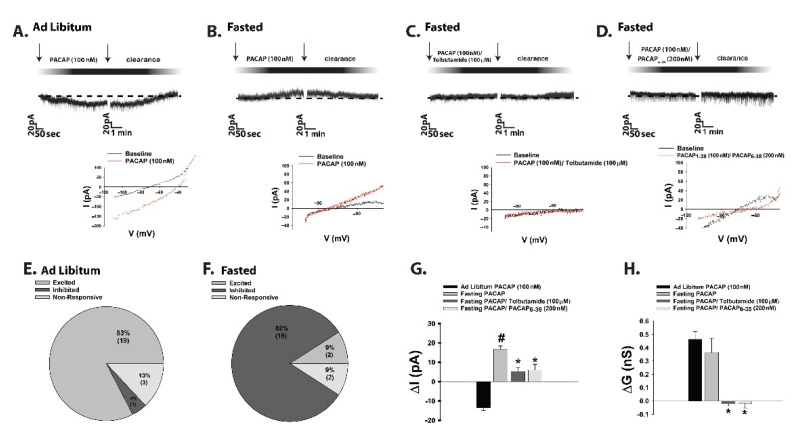
Fasting reverses the polarity of postsynaptic PACAP response in POMC neurons by switching the coupling of PAC1 receptors from TRPC5 to K_ATP_ channels. (**A**–**D**), Representative current traces from voltage clamp recordings of POMC neurons that depict the response elicited by 100 nM PACAP under ad libitum and fasting conditions, the latter of which is abrogated by blockade of K_ATP_ channels with tolbutamide (100 μM) and PAC1 receptors with PACAP_6–38_ (200 nM). The inset I/V plots illustrate that the fasting-induced change in polarity is due to a switch from a mixed cation conductance to a PAC1 receptor-mediated K^+^ conductance via K_ATP_ channels. (**E**,**F**), Pie charts that show the proportion of POMC neurons that are excited by, inhibited by, or unresponsive to PACAP under ad libitum (to the left) and fasting (to the right) conditions. (**G**,**H**) highlight the PACAP-induced changes in membrane current (ΔI) and conductance (ΔG); alone (*n* = 12) and in conjunction with tolbutamide (*n* = 12) and PACAP_6–38_ (*n* = 8). Bars represent means and lines 1 SEM. #, *p* < 0.05 relative to PACAP under ad libitum conditions. *, *p* < 0.05 relative to PACAP under fasting conditions, one-way ANOVA/LSD. Figure adapted from [12].

**Figure 3 ijms-22-02728-f003:**
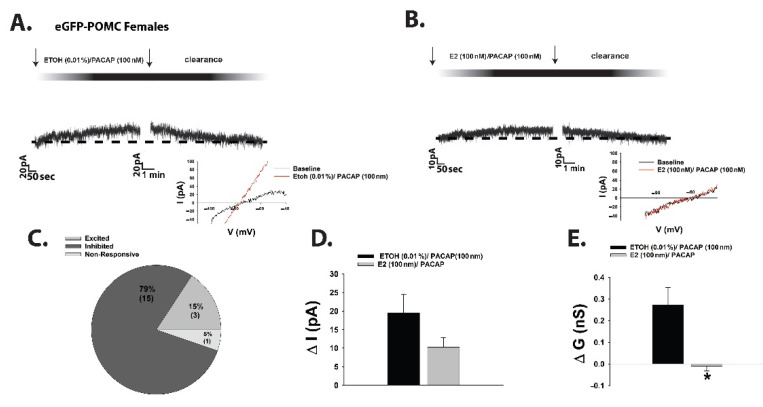
E_2_ attenuates the PACAP-induced outward current in POMC neurons observed under fasting conditions. (**A**,**B**) are representative membrane current traces during voltage clamp recordings in EtOH vehicle- (0.01% (*v:v*); *n* = 11) and E_2_-treated (100 nM; *n* = 8) slices from OVX females that illustrate the estrogenic diminution of the robust and reversible PACAP-induced outward current and change in K^+^ conductance (as seen from the inset I/V plots). (**C**), Pie chart that indicates the percentage of POMC neurons from OVX females that are excited by, inhibited by, or unresponsive to PACAP under fasting conditions. (**D**,**E**) show the composite data underscoring the ability of E_2_ to negatively modulate the PACAP response. Bars represent means and lines 1 SEM of the PACAP-induced ΔI and ΔG in POMC neurons from OVX females under fasting conditions. *, *p* < 0.05 relative to EtOH vehicle, Student’s *t*-test.

**Figure 4 ijms-22-02728-f004:**
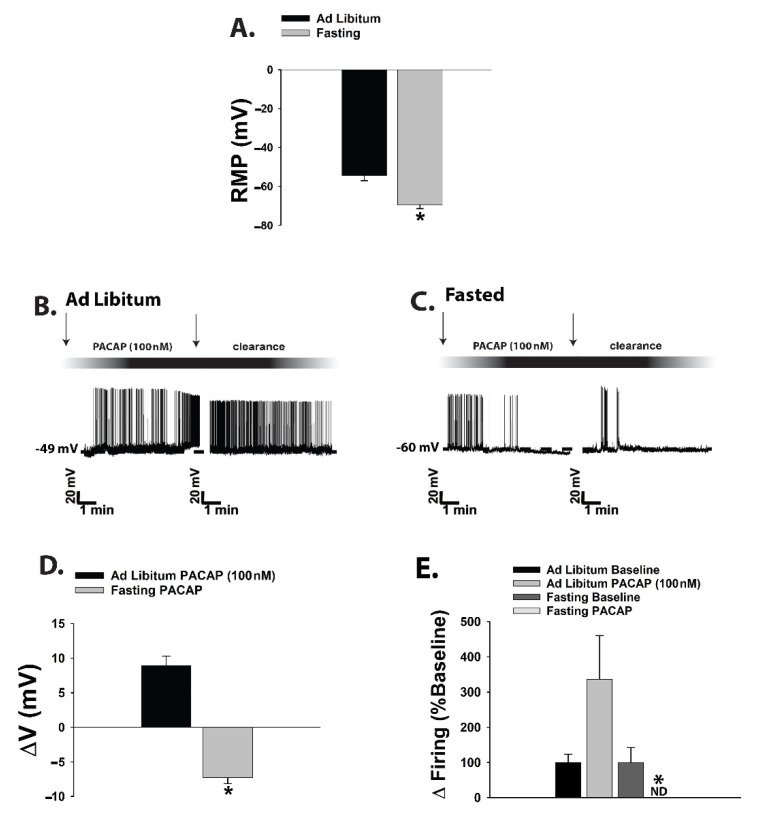
The PACAP-induced outward current observed during fasting conditions is associated with a hyperpolarization and a decrease in firing. (**A**), Composite bar graph that demonstrates the more hyperpolarized RMP of POMC neurons under fasting conditions. (**B**,**C**), Representative current clamp traces from POMC neurons showing the reversible PACAP-induced depolarization and increase in firing under ad libitum-fed conditions (*n* = 10) and the reversible hyperpolarization and suppression of firing seen under fasting conditions (*n* = 10). Comparable effects are seen during recordings in vehicle pre-treated slices from OVX females. (**D**,**E**), Composite data illustrating the PACAP-induced changes in membrane potential (ΔV) and firing rate under ad libitum-fed and fasting conditions. Bars represent means and lines 1 SEM. * *p* < 0.05, relative to ad libitum-fed conditions, Student’s *t*-test (**D**); relative to baseline, Kruskal–Wallis/median-notched box-and-whisker analysis (**E**). Figure adapted from [12].

**Figure 5 ijms-22-02728-f005:**
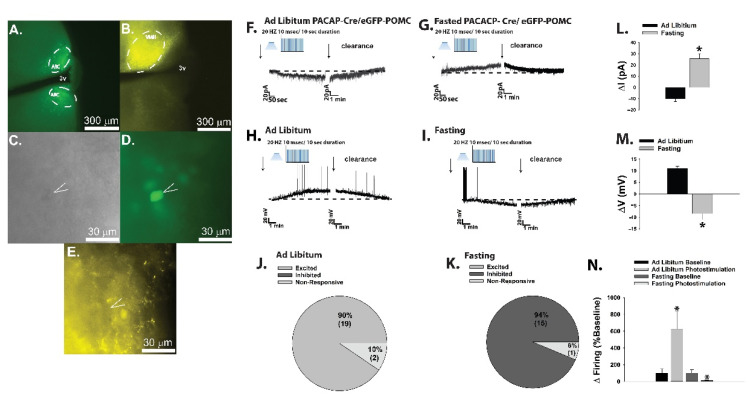
Optogenetic stimulation of VMN PACAP neurons depolarizes POMC neurons and increases their firing under ad libitum-fed conditions, effects which are flipped under fasting conditions. (**A**)**,** Low power (4×) image of ARC POMC neurons taken from a PACAP-Cre/eGFP POMC mouse. (**B**), Photomicrograph (4×) showing the channel rhodopsin-2 expression in VMN PACAP neurons two weeks after AAV injection as visualized by eYFP. (C), Differential interference contrast image (40×) of a recorded POMC neuron and the corresponding eGFP fluorescence signal from the same neuron (**D**). (**E**), 40X image showing the eYFP-labeled fibers in the immediate vicinity of the recorded neuron. Photostimulation (10-ms pulses delivered at 20 Hz for 10 s) of male VMN PACAP neurons produces a reversible inward current linked to membrane depolarization and increase in the firing of ARC POMC under ad libitum-fed conditions neurons (**F**,**H**,**J**,**L**–**N)**; *n* = 7–11), and the exact opposite is seen under fasting conditions **(G**,**I**,**K**–**N**); *n* = 11–13**)**. Bars represent means, and lines 1 SEM of the light-induced change in ΔI (**L**), ΔV (**M**) and firing rate (**N**). * *p* < 0.05, relative to ad libitum-fed conditions, Student’s *t*-test (**L**,**M**); relative to baseline, Kruskal–Wallis/median-notched box-and-whisker analysis (**N**). Figure adapted from [12].

**Figure 6 ijms-22-02728-f006:**
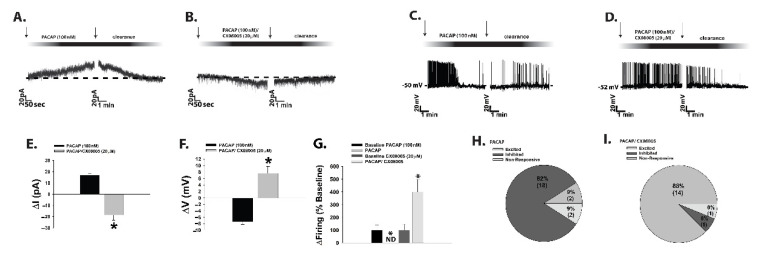
The fasting-induced switch in the polarity of the PACAP response in POMC neurons involves the activation of protein tyrosine phosphatases. Representative traces show that the PACAP-induced outward current (**A**; *n* = 12) and hyperpolarization (**C**; *n* = 10) seen under fasting conditions reverts back to an inward current (**B**; *n* = 9) and depolarization (**D**; *n* = 7) in the presence of the PTP1B/TCPTP inhibitor CX08005 (20 μM; **B**). This is further substantiated by the composite bar graphs in (**E**–**G**) as well as the pie charts in (**H**,**I**). Bars represent means, and lines 1 SEM of the PACAP-induced change ΔI, ΔV or normalized firing rate under fasted conditions, alone and in combination with CX08005, Compound C or metformin. * *p* < 0.05, relative to PACAP alone, Student’s *t*-test (**E****,F**); relative to baseline, Kruskal–Wallis/median-notched box-and-whisker analysis (**G**).

**Figure 7 ijms-22-02728-f007:**
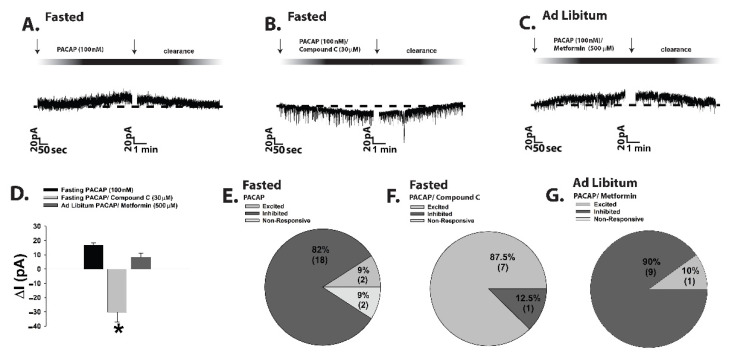
The fasting-induced reversal of the PACAP response in POMC neurons is also dependent upon activation of AMPK. The outward current caused by PACAP under fasting conditions in POMC neurons (**A**; *n* = 12) is transformed into an inward current in the presence of the AMPK inhibitor Compound C (30 μM; **B**; *n* = 8) The PACAP-induced outward current in (**A**) was reproduced under ad libitum-fed conditions in the presence of the AMPK activator metformin (500 μM; **C**; *n* = 10). The data from these representative traces is summarized in composite form by the bar graph in (**D**) and the pie charts in (**E**–**G**). Bars represent means and lines 1 SEM. *, *p* < 0.05 relative to PACAP alone, one-way ANOVA/LSD.

**Figure 8 ijms-22-02728-f008:**
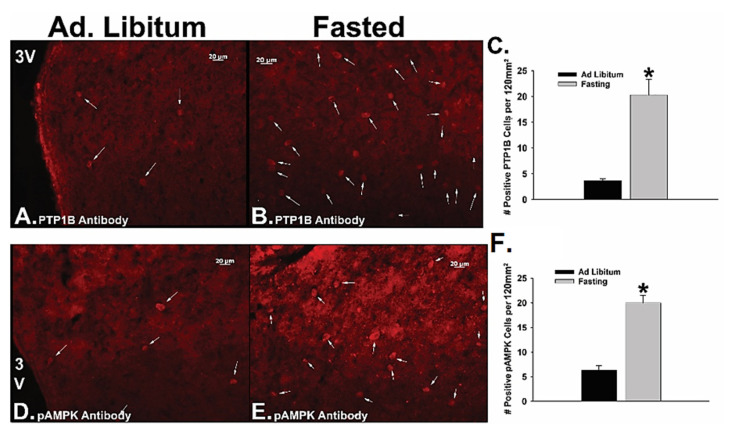
Fasting augments the activity and expression of PTP1B and AMPK in the ARC. The four 20× images depict the PTP1B (**A**,**B**; 1:100) and pAMPK (**D**,**E**; 1:100) immunoreactivity under ad libitum-fed (**A**,**D**) and fasting (**B**,**E**) as visualized with AF546 (1:600). The composite data in the bar graphs summarize the fasting-induced increase in the number of PTP1B- (**C**) and pAMPK-immunoreactive (**F**) cells per capita in the ARC. Bars represent means and lines 1 SEM. * *p* < 0.05, relative to ad libitum-fed conditions, Student’s *t*-test.

**Figure 9 ijms-22-02728-f009:**
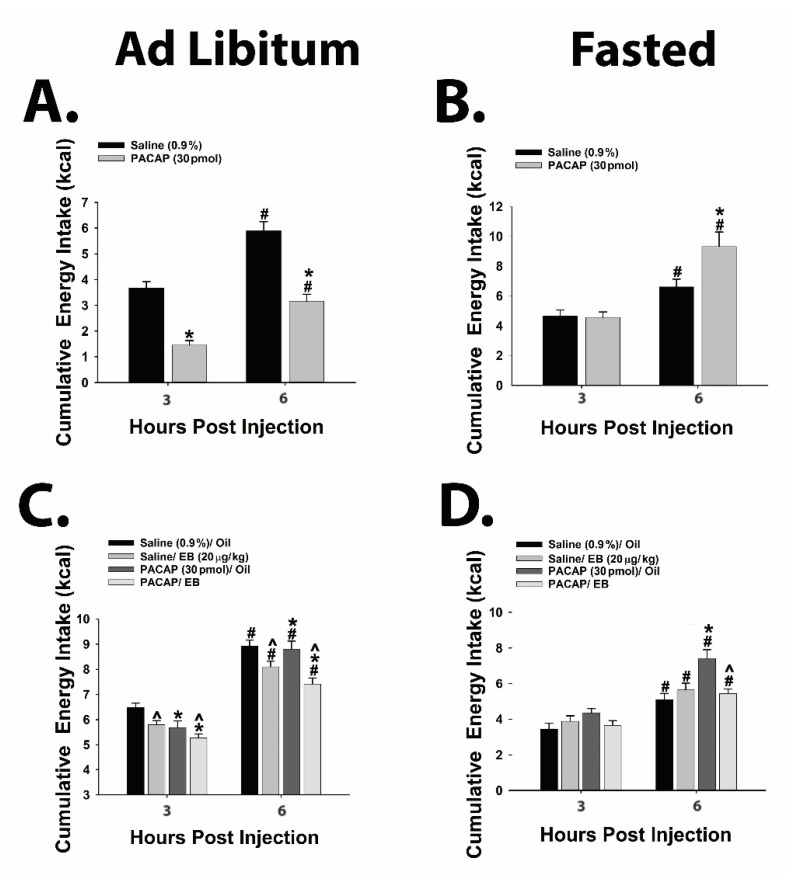
Fasting completely reverses the effect of a direct injection of PACAP into the ARC on energy intake, which is attenuated by EB in OVX females. Under ad libitum-fed conditions, PACAP (30pmo; *n* = 6l) significantly decreases cumulative energy intake in wildtype males compared to saline-treated controls (0.2 μL; **A**; *n* = 6). This PACAP-induced decrease in energy intake is no longer apparent in fasted males, and PACAP actually causes an increase in cumulative consumption which is evident at six hours post-administration (**B**; *n* = 7–9). PACAP also decreases energy intake in ad libitum-fed OVX wildtype females, and this effect is potentiated by EB (20 μg/kg; s.c.; **C**; *n* = 6). Again, the effect of PACAP on consummatory behavior in fasted OVX females is exactly the opposite of that seen under ad libitum conditions, as is the modulatory effect of EB (**D**; *n* = 6). Bars represent means and lines 1 S.E.M. of the cumulative energy intake seen in ad libitum-fed or fasted mice injected with either PACAP or its saline vehicle. #, *p* < 0.05 relative to cumulative energy intake seen at three hours after PACAP injection, repeated-measures, multi-factorial ANOVA/LSD; *, *p* < 0.05 relative to saline vehicle, repeated measures, multi-factorial ANOVA/LSD; ^, *p* < 0.05 relative to sesame oil vehicle, repeated measures, multi-factorial ANOVA/LSD. Figure adapted from [12].

**Figure 10 ijms-22-02728-f010:**
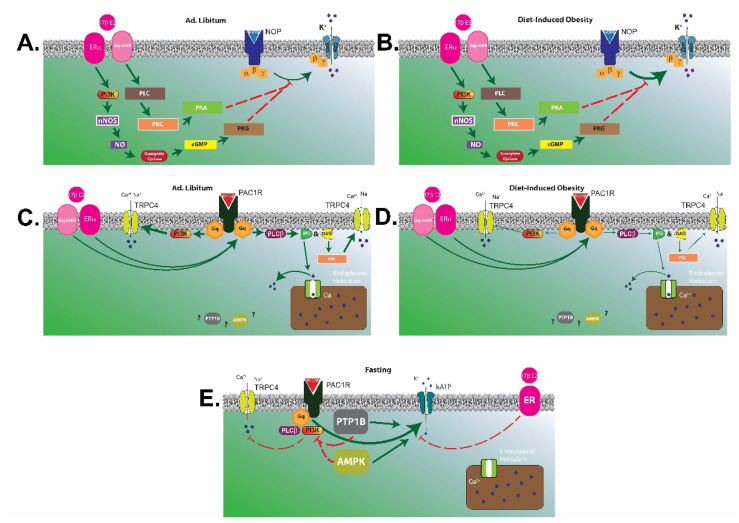
Schematic diagrams illustrating how PAC1 and NOP receptor signaling in POMC neurons is altered under various energy balance states. (**A**), In POMC neurons from ad libitum-fed animals, N/OFQ activates the NOP receptor that initiates G_i/o_-mediated signaling and subsequent activation of GIRK channels through positive allosteric modification by the βγ complex. This in turn promotes K^+^ efflux and inhibition of POMC neurons, effects which are dampened by E_2_ acting through ERα and G_q_-mER to stimulate PI3K and nNOS as well as PLC, PKC and PKA signaling pathways, respectively. (**B**), In POMC neurons from obese animals, NOP receptor/effector coupling is enhanced; leading to a greater inhibitory effect of N/OFQ on POMC neurons. This N/OFQ-induced inhibition of POMC neurons is once again abrogated by E_2_ in POMC neurons from obese females. (**C**), Under ad libitum conditions, PACAP activates its cognate PAC1 receptor to elicit G_q_-mediated signaling; working through PI3K as well as PLC, IP3, DAG and PKC to promote Ca^2+^ mobilization from intracellular stores and the coupling of PAC1 receptors to TRCP5 channels. This leads to cation flux through the channel pore that depolarizes and thereby excites POMC neurons. In females, E_2_ can act via ERα and Gq-mER to potentiate PAC1 receptor/TRPC5 channel coupling and PACAP-induced excitation of POMC neurons. (**D**), Under conditions of diet-induced obesity, the PAC1 receptor-mediated activation of TRPC5 channels in male POMC neurons is attenuated. However, in obese females the PACAP-induced excitation of POMC neurons is maintained due to the potentiating effect of E_2_. (**E**), Under conditions of fasting, the expression and activity of AMPK and protein tyrosine phosphatases like PTP1B is elevated in POMC neurons. This triggers a switch in the coupling of PAC1 receptors, such that they now are no longer linked with TRPC5 channels and instead inhibit rather than excite POMC neurons via activation of K_ATP_ channels. This inhibitory effect of PACAP in POMC neurons from fasted animals is diminished by E_2_ in females.

## Data Availability

The data presented in this study are available in Gastelum, C.; Perez, L.; Hernandez, J.; Le, N.; Vahrson, I.; Sayers, S.; Wagner, E.J. Adaptive Changes in the Central Control of Energy Homeostasis Occur in Response to Variations in Energy Status. Int. J. Mol. Sci. 2021, 22, 2728. doi.org/10.3390/ijms22052728.

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
