# Peer review of "Adaptive Changes in the Central Control of Energy Homeostasis Occur in Response to Variations in Energy Status"

_ijms, 2021, doi:10.3390/ijms22052728_

Round 1

Reviewer 1 Report

In this manuscript, Gastelum et al. discussed the roles of FQ and PACAP in control of energy homeostasis. overall, it is a well-organized and comprehensively described review article that significantly contribute to the field. I only have a few suggestions.

1. The title is too complicated. The authors should make it shorter and clearer.

2. The authors should discuss more deeply on the aspect of sex differences, in particular, the specific sex hormones.

3. The authors should pay more attention to the splice variants while discussing NOS1, especially, regarding to the organs expressing NOS1b but not NOS1a as primary isoform.

Author Response

Reviewer #1

1) The title is too complicated. The authors should make it shorter and clearer.

We have changed the title of the manuscript in accordance with the reviewer’s suggestion.

2) The authors should discuss more deeply on the aspect of sex differences, in particular, the specific sex hormones.

We appreciate that the reviewer recognizes the importance of sex differences as well as the activational effects of gonadal hormones in the homeostatic and hedonic regulation of energy balance.  We share this recognition, and this is precisely why we devoted so much time and effort on the topic in our original submission.  We discussed sex differences in, and gonadal hormonal influences on, the cannabinoid regulation of homeostatic energy balance (page 8, lines 336-349; page 9, lines 360-366), the development of central insulin resistance (page 9, lines 366-371), the NOP receptor-mediated regulation of the homeostatic (page 11, lines 463-481, 489-491) and hedonic (page 12, lines 519-536) energy balance circuitries, and finally the PAC1 receptor-mediated regulation of energy balance (page 15, lines 669-691; Page 15-22, lines 701-878).  The discussion of sex differences in NOP and PAC1 receptor-mediated regulation of energy homeostasis was included in the sections specific to N/OFQ and PACAP.  Thus, as hopefully the reviewer now appreciates, we did in fact provide significant coverage to the topic of sex differences in the central regulation of energy homeostasis. 

3) The authors should pay more attention to the splice variants while discussing NOS1, especially, regarding to the organs expressing NOS1b but not NOS1a as primary isoform.

Our intent was to introduce nNOS as an important signaling molecule in the negative modulatory crosstalk between ERα and orexigenic cannabinoid CB1 receptors as well as NOP receptors involved in the central regulation of energy homeostasis.  However, the western blotting and pharmacological means by which we determined this did not allow us to distinguish between the different isoforms.  A statement to this effect has been provided on page 8, lines 345-349 of the revised manuscript.

Reviewer 2 Report

This manuscript by Gastelum and colleagues addresses the regulation of energy homeostasis by the hypothalamus and mesolimbic dopamine pathway, with a special focus on nociceptin/orphanin FQ (N/OFQ) and pituitary adenylyate cyclase-activating polypeptide (PACAP). The manuscript begins with a review of homeostatic and hedonic energy balance regulation, identifying neural, hormonal, and humoral factors that participate in these phenomena. It then reviews the role of the N/OFQ system in homeostatic and hedonic energy balance regulation. Finally, it reviews the role of PACAP in these phenomena and presents new data that support the idea that the anorexigenic effect of PACAP and its excitation of POMC neurons in ad libitum fed rats switches to an orexigenic effect and inhibition of POMC neurons in fasted rats.

The review of energy regulation covers a vast field and the new data add significantly to it. The final schematic presents an excellent summary of how N/OFQ and PACAP signaling in POMC neurons is altered under several energy balance states. There are, however, several issues that must be addressed before this manuscript can be acceptable for publication.

Major Issues:

  1. The fasting results presented in the last several pages of the manuscript appear to be new (previously unpublished), but this is not immediately clear. The data that are new here should be clearly delineated. Full statistics should also be included for these results.

  1. While the text in each paragraph is consistent in its theme, several paragraphs lack well-defined structure, making their central argument difficult to determine. Clear thesis and concluding sentences (and perhaps breaking up of some of the paragraphs) will help the main messages to become more explicit and help readers to more easily follow the authors’ logic.

  1. It would be useful to include a schematic for Section 1. The authors seem to have a clear vision of how the hypothalamic factors work together; having a schematic here would make this easier for the reader to grasp.

  1. The discussion about dopamine encoding of reward processing neglects significant recent evidence that also implicates accumbal dopamine in aversion processing (see, for example, Soares-Cunha et al, 2020 Mol Psychiatry; Yuan et al, 2019 J Neuro; de Jong et al, 2019 Neuron, etc). The authors should take a more balanced approach in their consideration of the role of dopamine in energy balance.

  1. The findings that the nociceptin/ Orphanin FQ system is largely hyperphagic but reduces dopamine system activity appears to contradict the authors’ stance that dopamine acts to promote food seeking. This apparent inconsistency should be reconciled.

Minor Issues:

  1. The full name of PACAP is misspelled as “adenylyl” instead of “adenylate” throughout the manuscript, in the title (line 4), abstract (line 21), keywords (line 28), and main text (lines 491, 493).

  1. On first mention, fa/fa rats and ob/ob mice should be briefly described.

  1. The term “addicts” (line 255) is stigmatizing. Instead, please use “people with substance use disorders” or some other similar terminology.

  1. Please explain Type II receptors (line 516).

Author Response

Reviewer #2

1) The fasting results presented in the last several pages of the manuscript appear to be new (previously unpublished), but this is not immediately clear. The data that are new here should be clearly delineated. Full statistics should also be included for these results.

The reviewer is correct in that we did include some new fasting data as a follow-up to our full-length article detailing the anorexigenic effect of PACAP under ad libitum conditions that occurs at least in part via PAC1 receptor-mediated activation of TRPC5 channels and subsequent excitation of POMC neurons.  (Chang et al., Neuroendocrinology, 2020).  The materials and methods were included in the original submission as a supplemental file. Per the reviewer’s suggestion we have embedded the statistical readout for all the new data described in section 5 of the revised manuscript. 

2) While the text in each paragraph is consistent in its theme, several paragraphs lack well-defined structure, making their central argument difficult to determine. Clear thesis and concluding sentences (and perhaps breaking up of some of the paragraphs) will help the main messages to become more explicit and help readers to more easily follow the authors’ logic.

Per the reviewer’s recommendation we have added concluding sentences and broken up paragraphs at several junctures throughout the revised manuscript; particularly in Sections 3-5.

3) It would be useful to include a schematic for Section 1. The authors seem to have a clear vision of how the hypothalamic factors work together; having a schematic here would make this easier for the reader to grasp.

Per the reviewers request we have included a schematic (Figure 1 of the revised manuscript) that captures the interplay between peripheral hormones and the homeostatic energy balance circuitry described in Section 1.

4) The discussion about dopamine encoding of reward processing neglects significant recent evidence that also implicates accumbal dopamine in aversion processing (see, for example, Soares-Cunha et al, 2020 Mol Psychiatry; Yuan et al, 2019 J Neuro; de Jong et al, 2019 Neuron, etc). The authors should take a more balanced approach in their consideration of the role of dopamine in energy balance.

The 254 references cited in the original submission are indicative of a thorough and balanced coverage of the hedonic energy balance circuitry described in Section 2.  Recognizing that you simply cannot catch everything out there in the literature, we certainly appreciate the recommended articles and will include them as part of a paragraph devoted to the role of accumbens dopamine in aversion processing.

5) The findings that the nociceptin/ Orphanin FQ system is largely hyperphagic but reduces dopamine system activity appears to contradict the authors’ stance that dopamine acts to promote food seeking. This apparent inconsistency should be reconciled.

To clarify, despite the inhibitory effect on the hedonic energy balance circuitry there net hyperphagic response caused by N/OFQ that is elicited, for example, by icv administration of the peptide. This is due largely, if not exclusively, to its effects within the homeostatic energy balance circuitry.  A statement to this effect has been added to the end of Section 4 (page 12, line 536-540) of the revised manuscript.

Minor issues:

1) The full name of PACAP is misspelled as “adenylyl” instead of “adenylate” throughout the manuscript, in the title (line 4), abstract (line 21), keywords (line 28), and main text (lines 491, 493).

We have corrected this in the revised manuscript.

2) On first mention, fa/fa rats and ob/ob mice should be briefly described.

Done.

3) The term “addicts” (line 255) is stigmatizing. Instead, please use “people with substance use disorders” or some other similar terminology.

This has been remedied. 

4) Please explain Type II receptors (line 516).

Done. 

Round 2

Reviewer 2 Report

The authors have addressed all of my major concerns.

There are still two places where the text should be changed from "adenylyl" to "adenylate" (lines 45 and 541), but this can be corrected at the copy editing stage.